# Localized Data Fusion for Kernel $k$-Means Clustering with Application to Cancer Biology

**Mehmet Gönen**
gonen@ohsu.edu
Department of Biomedical Engineering
Oregon Health & Science University
Portland, OR 97239, USA

**Adam A. Margolin**
margolin@ohsu.edu
Department of Biomedical Engineering
Oregon Health & Science University
Portland, OR 97239, USA

## Abstract

In many modern applications from, for example, bioinformatics and computer vision, samples have multiple feature representations coming from different data sources. Multiview learning algorithms try to exploit all these available information to obtain a better learner in such scenarios. In this paper, we propose a novel multiple kernel learning algorithm that extends kernel $k$-means clustering to the multiview setting, which combines kernels calculated on the views in a localized way to better capture sample-specific characteristics of the data. We demonstrate the better performance of our localized data fusion approach on a human colon and rectal cancer data set by clustering patients. Our method finds more relevant prognostic patient groups than global data fusion methods when we evaluate the results with respect to three commonly used clinical biomarkers.

## 1 Introduction

Clustering algorithms aim to find a meaningful grouping of the samples at hand in an unsupervised manner for exploratory data analysis. $k$-means clustering is one of the classical algorithms (Hartigan, 1975), which uses $k$ prototype vectors (i.e., centers or centroids of $k$ clusters) to characterize the data and minimizes a sum-of-squares cost function to find these prototypes with a coordinate descent optimization method. However, the final cluster structure heavily depends on the initialization because the optimization scheme of $k$-means clustering is prone to local minima. Fortunately, the sum-of-squares minimization can be formulated as a trace maximization problem, which can not be solved easily due to binary decision variables used to denote cluster memberships, but this hard optimization problem can be reduced to an eigenvalue decomposition problem by relaxing the constraints (Zha et al., 2001; Ding and He, 2004). In such a case, overall clustering algorithm can be formulated in two steps: (i) performing principal component analysis (PCA) (Pearson, 1901) on the covariance matrix and (ii) recovering cluster membership matrix using the $k$ eigenvectors that correspond to the $k$ largest eigenvalues. Similar to many other learning algorithms, $k$-means clustering is also extended towards a nonlinear version with the help of kernel functions, which is called kernel $k$-means clustering (Girolami, 2002). The kernelized variant can also be optimized with a spectral relaxation approach using kernel PCA (KPCA) (Schölkopf et al., 1998) instead of canonical PCA.

In many modern applications, samples have multiple feature representations (i.e., views) coming from different data sources. Instead of using only one of the views, it is better to use all available information and let the learning algorithm decide how to combine these data sources, which is known as multiview learning. There are three main categories for the combination strategy (Noble, 2004): (i) combination at the feature level by concatenating the views (i.e., early integration), (ii) combination at the decision level by concatenating the outputs of learners trained on each view separately (i.e., late integration), and (iii) combination at the learning level by trying to find a unified distance, kernel, or latent matrix using all views simultaneously (i.e., intermediate integration).

## 1.1 Related work

When we have multiple views for clustering, we can simply concatenate the views and train a standard clustering algorithm on the concatenated view, which is known as early integration. However, this approach does not assign weights to the views, and the view with the highest number of features might dominate the clustering step due to the unsupervised nature of the problem.

Late integration algorithms obtain a clustering on each view separately and combine these clustering results using an ensemble learning scheme. Such clustering algorithms are also known as cluster ensembles (Strehl and Ghosh, 2002). However, they do not exploit the dependencies between the views during clustering, and these dependencies might already be lost if we combine only clustering results in the second step.

Intermediate integration algorithms combine the views in a single learning scheme to collectively find a unified clustering. Chaudhuri et al. (2009) propose to extract a unifying feature representation from the views by performing canonical correlation analysis (CCA) (Hotelling, 1936) and to train a clustering algorithm on this common representation. Similarly, Blaschko and Lampert (2008) extract a common feature representation but with a nonlinear projection step using kernel CCA (Lai and Fyfe, 2000) and then perform clustering. Such CCA-based algorithms assume that all views are informative, and if there are some noisy views, this can degrade the clustering performance drastically. Lange and Buhmann (2006) propose to optimize the weights of a convex combination of view-specific similarity measures within a nonnegative matrix factorization framework and to assign samples to clusters using the latent matrices obtained in the factorization step. Valizadegan and Jin (2007) extend the maximum margin clustering formulation of Xu et al. (2004) to perform kernel combination and clustering jointly by formulating a semidefinite programming (SDP) problem. Chen et al. (2007) further improve this idea by formulating a quadratically constrained quadratic programming problem instead of an SDP problem. Tang et al. (2009) convert the views into graphs by placing samples into vertices and creating edges using the similarity values between samples in each view, and then factorize these graphs jointly with a shared factor common to all graphs, which is used for clustering at the end. Kumar et al. (2011) propose a co-regularization strategy for multiview spectral clustering by enforcing agreement between the similarity matrices calculated on the latent representations obtained from the spectral decomposition of each view. Huang et al. (2012) formulate another multiview spectral clustering method that finds a weighted combination of the affinity matrices calculated on the views. Yu et al. (2012) develop a multiple kernel $k$-means clustering algorithm that optimizes the weights in a conic sum of kernels calculated on the views. However, their formulation uses the same kernel weights for all of the samples.

Multiview clustering algorithms have attracted great interest in cancer biology due to the availability of multiple genomic characterizations of cancer patients. Yuan et al. (2011) formulate a patient-specific data fusion algorithm that uses a nonparametric Bayesian model coupled with a Markov chain Monte Carlo inference scheme, which can combine only two views and is computationally very demanding due to the high dimensionality of genomic data. Shen et al. (2012) and Mo et al. (2013) find a shared latent subspace across genomic views and cluster cancer patients using their representations in this subspace. Wang et al. (2014) construct patient networks from patient–patient similarity matrices calculated on the views, combine these into a single unified network using a network fusion approach, and then perform clustering on the final patient network.

## 1.2 Our contributions

Intermediate integration using kernel matrices is also known as multiple kernel learning (MKL) (Gönen and Alpaydın, 2011). Most of the existing MKL algorithms use the same kernel weights for all samples, which may not be a good idea due to sample-specific characteristics of the data or measurement noise present in some of the views. In this work, we study kernel $k$-means clustering under the multiview setting and propose a novel MKL algorithm that combines kernels with sample-specific weights to obtain a better clustering. We demonstrate the better performance of our algorithm on the human colon and rectal cancer data set provided by TCGA consortium (The Cancer Genome Atlas Network, 2012), where we use three genomic characterizations of the patients (i.e., DNA copy number, mRNA gene expression, and DNA methylation) for clustering. Our localized data fusion approach obtains more relevant prognostic patient groups than global fusion approaches when we evaluate the results with respect to three commonly used clinical biomarkers (i.e., microsatellite instability, hypermutation, and mutation in BRAF gene) of colon and rectal cancer.

## 2 Kernel $k$-means clustering

We first review kernel $k$-means clustering (Girolami, 2002) before extending it to the multiview setting. Given $N$ independent and identically distributed samples $\{\boldsymbol{x}_i \in \mathcal{X}\}_{i=1}^n$, we assume that there is a function $\Phi(\cdot)$ that maps the samples into a feature space, in which we try to minimize a sum-of-squares cost function over the cluster assignment variables $\{z_{ic}\}_{i=1,c=1}^{n,k}$. The optimization problem (OPT1) defines kernel $k$-means clustering as a binary integer programming problem, where $n_c$ is the number of samples assigned to cluster $c$, and $\boldsymbol{\mu}_c$ is the centroid of cluster $c$.

$$
\begin{aligned}
\text{minimize} \quad & \sum_{i=1}^n \sum_{c=1}^k z_{ic} \|\Phi(\boldsymbol{x}_i) - \boldsymbol{\mu}_c\|_2^2 \\
\text{with respect to} \quad & z_{ic} \in \{0,1\} \quad \forall (i,c) \\
\text{subject to} \quad & \sum_{c=1}^k z_{ic} = 1 \quad \forall i \\
\text{where} \quad & n_c = \sum_{i=1}^n z_{ic} \quad \forall c, \quad \boldsymbol{\mu}_c = \frac{1}{n_c} \sum_{i=1}^n z_{ic} \Phi(\boldsymbol{x}_i) \quad \forall c
\end{aligned}
\tag{OPT1}
$$

We can convert this optimization problem into an equivalent matrix-vector form problem as follows:

$$
\begin{aligned}
\text{minimize} \quad & \operatorname{tr}\left((\boldsymbol{\Phi} - \mathbf{M})^\top (\boldsymbol{\Phi} - \mathbf{M})\right) \\
\text{with respect to} \quad & \mathbf{Z} \in \{0,1\}^{n \times k} \\
\text{subject to} \quad & \mathbf{Z}\mathbf{1}_k = \mathbf{1}_n \\
\text{where} \quad & \boldsymbol{\Phi} = [\Phi(\boldsymbol{x}_1) \quad \Phi(\boldsymbol{x}_2) \quad \ldots \quad \Phi(\boldsymbol{x}_n)], \quad \mathbf{M} = \boldsymbol{\Phi}\mathbf{Z}\mathbf{L}\mathbf{Z}^\top, \\
& \mathbf{L} = \operatorname{diag}(n_1^{-1}, n_2^{-1}, \ldots, n_k^{-1}).
\end{aligned}
\tag{OPT2}
$$

Using that $\boldsymbol{\Phi}^\top \boldsymbol{\Phi} = \mathbf{K}$, $\operatorname{tr}(\mathbf{AB}) = \operatorname{tr}(\mathbf{BA})$, and $\mathbf{Z}^\top \mathbf{Z} = \mathbf{L}^{-1}$, the objective function of the optimization problem (OPT2) can be rewritten as

$$
\begin{aligned}
\operatorname{tr}\left((\boldsymbol{\Phi} - \mathbf{M})^\top (\boldsymbol{\Phi} - \mathbf{M})\right) &= \operatorname{tr}\left((\boldsymbol{\Phi} - \boldsymbol{\Phi}\mathbf{Z}\mathbf{L}\mathbf{Z}^\top)^\top (\boldsymbol{\Phi} - \boldsymbol{\Phi}\mathbf{Z}\mathbf{L}\mathbf{Z}^\top)\right) \\
&= \operatorname{tr}\left(\boldsymbol{\Phi}^\top \boldsymbol{\Phi} - 2\boldsymbol{\Phi}^\top \boldsymbol{\Phi}\mathbf{Z}\mathbf{L}\mathbf{Z}^\top + \mathbf{Z}\mathbf{L}\mathbf{Z}^\top \boldsymbol{\Phi}^\top \boldsymbol{\Phi}\mathbf{Z}\mathbf{L}\mathbf{Z}^\top\right) \\
&= \operatorname{tr}\left(\mathbf{K} - 2\mathbf{K}\mathbf{Z}\mathbf{L}\mathbf{Z}^\top + \mathbf{K}\mathbf{Z}\mathbf{L}\mathbf{Z}^\top \mathbf{Z}\mathbf{L}\mathbf{Z}^\top\right) = \operatorname{tr}\left(\mathbf{K} - \mathbf{L}^{\frac{1}{2}}\mathbf{Z}^\top \mathbf{K}\mathbf{Z}\mathbf{L}^{\frac{1}{2}}\right),
\end{aligned}
$$

where $\mathbf{K}$ is the kernel matrix that holds the similarity values between the samples, and $\mathbf{L}^{\frac{1}{2}}$ is defined as taking the square root of the diagonal elements. The resulting optimization problem (OPT3) is a trace maximization problem, but it is still very difficult to solve due to the binary decision variables.

$$
\begin{aligned}
\text{maximize} \quad & \operatorname{tr}\left(\mathbf{L}^{\frac{1}{2}}\mathbf{Z}^\top \mathbf{K}\mathbf{Z}\mathbf{L}^{\frac{1}{2}} - \mathbf{K}\right) \\
\text{with respect to} \quad & \mathbf{Z} \in \{0,1\}^{n \times k} \\
\text{subject to} \quad & \mathbf{Z}\mathbf{1}_k = \mathbf{1}_n
\end{aligned}
\tag{OPT3}
$$

However, we can formulate a relaxed version of this optimization problem by renaming $\mathbf{Z}\mathbf{L}^{\frac{1}{2}}$ as $\mathbf{H}$ and letting $\mathbf{H}$ take arbitrary real values subject to orthogonality constraints.

$$
\begin{aligned}
\text{maximize} \quad & \operatorname{tr}\left(\mathbf{H}^\top \mathbf{K}\mathbf{H} - \mathbf{K}\right) \\
\text{with respect to} \quad & \mathbf{H} \in \mathbb{R}^{n \times k} \\
\text{subject to} \quad & \mathbf{H}^\top \mathbf{H} = \mathbf{I}_k
\end{aligned}
\tag{OPT4}
$$

The final optimization problem (OPT4) can be solved by performing KPCA on the kernel matrix $\mathbf{K}$ and setting $\mathbf{H}$ to the $k$ eigenvectors that correspond to $k$ largest eigenvalues (Schölkopf et al., 1998). We can finally extract a clustering solution by first normalizing all rows of $\mathbf{H}$ to be on the unit sphere and then performing $k$-means clustering on this normalized matrix. Note that, after the normalization step, $\mathbf{H}$ contains $k$-dimensional representations of the samples on the unit sphere, and $k$-means is not very sensitive to initialization in such a case.

## 3   Multiple kernel $k$-means clustering

In a multiview learning scenario, we have multiple feature representations, where we assume that each representation has its own mapping function, i.e., $\{\Phi_m(\cdot)\}_{m=1}^{p}$. Instead of an unweighted combination of these views (i.e., simple concatenation), we can obtain a weighted mapping function by concatenating views using a convex sum (i.e., nonnegative weights that sum up to 1). This corresponds to replacing $\Phi(\boldsymbol{x}_i)$ with $\Phi_{\boldsymbol{\theta}}(\boldsymbol{x}_i) = \begin{bmatrix} \theta_1\Phi_1(\boldsymbol{x}_i)^{\top} & \theta_2\Phi_2(\boldsymbol{x}_i)^{\top} & \ldots & \theta_p\Phi_p(\boldsymbol{x}_i)^{\top} \end{bmatrix}^{\top}$, where $\boldsymbol{\theta} \in \mathbb{R}_{+}^{p}$ is the vector of kernel weights that we need to optimize during training. The kernel function defined over the weighted mapping function becomes

$$k_{\boldsymbol{\theta}}(\boldsymbol{x}_i, \boldsymbol{x}_j) = \langle \Phi_{\boldsymbol{\theta}}(\boldsymbol{x}_i), \Phi_{\boldsymbol{\theta}}(\boldsymbol{x}_j) \rangle = \sum_{m=1}^{p} \langle \theta_m \Phi_m(\boldsymbol{x}_i), \theta_m \Phi_m(\boldsymbol{x}_j) \rangle = \sum_{m=1}^{p} \theta_m^2 k_m(\boldsymbol{x}_i, \boldsymbol{x}_j),$$

where we combine kernel functions using a conic sum (i.e., nonnegative weights), which guarantees to have a positive semi-definite kernel function at the end. The optimization problem (OPT5) gives the trace maximization problem we need to solve.

$$\begin{aligned} \text{maximize} \quad & \text{tr}\left(\mathbf{H}^{\top}\mathbf{K}_{\boldsymbol{\theta}}\mathbf{H} - \mathbf{K}_{\boldsymbol{\theta}}\right) \\ \text{with respect to} \quad & \mathbf{H} \in \mathbb{R}^{n \times k}, \quad \boldsymbol{\theta} \in \mathbb{R}_{+}^{p} \\ \text{subject to} \quad & \mathbf{H}^{\top}\mathbf{H} = \mathbf{I}_k, \quad \boldsymbol{\theta}^{\top}\mathbf{1}_p = 1 \\ \text{where} \quad & \mathbf{K}_{\boldsymbol{\theta}} = \sum_{m=1}^{p} \theta_m^2 \mathbf{K}_m \end{aligned} \qquad \text{(OPT5)}$$

We solve this problem using a two-step alternating optimization strategy: (i) **Optimize H given $\theta$.** If we know the kernel weights (or initialize randomly in the first iteration), solving (OPT5) reduces to solving (OPT4) with the combined kernel matrix $\mathbf{K}_{\boldsymbol{\theta}}$, which requires performing KPCA on $\mathbf{K}_{\boldsymbol{\theta}}$. (ii) **Optimize $\theta$ given H.** If we know the eigenvectors from the first step, solving (OPT5) reduces to solving (OPT6), which is a convex quadratic programming (QP) problem with $p$ decision variables and one equality constraint, and is solvable with any standard QP solver up to a moderate number of kernels.

$$\begin{aligned} \text{minimize} \quad & \sum_{m=1}^{p} \theta_m^2 \, \text{tr}\left(\mathbf{K}_m - \mathbf{H}^{\top}\mathbf{K}_m\mathbf{H}\right) \\ \text{with respect to} \quad & \boldsymbol{\theta} \in \mathbb{R}_{+}^{p} \\ \text{subject to} \quad & \boldsymbol{\theta}^{\top}\mathbf{1}_p = 1 \end{aligned} \qquad \text{(OPT6)}$$

Note that using a convex combination of kernels in (OPT5) is not a viable option because if we set $\mathbf{K}_{\boldsymbol{\theta}}$ to $\sum_{m=1}^{p} \theta_m \mathbf{K}_m$, there would be a trivial solution to the trace maximization problem with a single active kernel and others with zero weights, which is also observed by Yu et al. (2012).

## 4   Localized multiple kernel $k$-means clustering

Instead of using the same kernel weights for all samples, we propose to use a localized data fusion approach by assigning sample-specific weights to kernels, which enables us to capture sample-specific characteristics of the data and to get rid of sample-specific noise that may be present in some of the views. In our localized combination approach, the mapping function is represented as $\Phi_{\boldsymbol{\Theta}}(\boldsymbol{x}_i) = \begin{bmatrix} \theta_{i1}\Phi_1(\boldsymbol{x}_i)^{\top} & \theta_{i2}\Phi_2(\boldsymbol{x}_i)^{\top} & \ldots & \theta_{ip}\Phi_p(\boldsymbol{x}_i)^{\top} \end{bmatrix}^{\top}$, where $\boldsymbol{\Theta} \in \mathbb{R}_{+}^{n \times p}$ is the matrix of sample-specific kernel weights, which are nonnegative and sum up to 1 for each sample (Gönen and Alpaydın, 2013). The locally combined kernel function can be written as

$$k_{\boldsymbol{\Theta}}(\boldsymbol{x}_i, \boldsymbol{x}_j) = \langle \Phi_{\boldsymbol{\Theta}}(\boldsymbol{x}_i), \Phi_{\boldsymbol{\Theta}}(\boldsymbol{x}_j) \rangle = \sum_{m=1}^{p} \langle \theta_{im}\Phi_m(\boldsymbol{x}_i), \theta_{jm}\Phi_m(\boldsymbol{x}_j) \rangle = \sum_{m=1}^{p} \theta_{im}\theta_{jm} k_m(\boldsymbol{x}_i, \boldsymbol{x}_j),$$

where we are guaranteed to have a positive semi-definite kernel function. The optimization problem (OPT7) gives the trace maximization problem with the locally combined kernel matrix, where $\boldsymbol{\theta}_m \in \mathbb{R}_{+}^{n}$ is the vector of kernel weights assigned to kernel $m$, and $\circ$ denotes the Hadamard product.

$$\text{maximize} \quad \text{tr}\left(\mathbf{H}^{\top}\mathbf{K}_{\boldsymbol{\Theta}}\mathbf{H} - \mathbf{K}_{\boldsymbol{\Theta}}\right)$$
$$\text{with respect to} \quad \mathbf{H} \in \mathbb{R}^{n \times k}, \quad \boldsymbol{\Theta} \in \mathbb{R}_{+}^{n \times p}$$
$$\text{subject to} \quad \mathbf{H}^{\top}\mathbf{H} = \mathbf{I}_{k}, \quad \boldsymbol{\Theta}\mathbf{1}_{p} = \mathbf{1}_{n} \qquad \text{(OPT7)}$$
$$\text{where} \quad \mathbf{K}_{\boldsymbol{\Theta}} = \sum_{m=1}^{p}(\boldsymbol{\theta}_{m}\boldsymbol{\theta}_{m}^{\top}) \circ \mathbf{K}_{m}$$

We solve this problem using a two-step alternating optimization strategy: (i) **Optimize H given $\boldsymbol{\Theta}$.** If we know the sample-specific kernel weights (or initialize randomly in the first iteration), solving (OPT7) reduces to solving (OPT4) with the combined kernel matrix $\mathbf{K}_{\boldsymbol{\Theta}}$, which requires performing KPCA on $\mathbf{K}_{\boldsymbol{\Theta}}$. (ii) **Optimize $\boldsymbol{\Theta}$ given H.** If we know the eigenvectors from the first step, using that $\text{tr}\left(\mathbf{A}^{\top}((\boldsymbol{cc}^{\top}) \circ \mathbf{B})\mathbf{A}\right) = \boldsymbol{c}^{\top}((\mathbf{A}\mathbf{A}^{\top}) \circ \mathbf{B})\boldsymbol{c}$, solving (OPT7) reduces to solving (OPT8), which is a convex QP problem with $n \times p$ decision variables and $n$ equality constraints.

$$\text{minimize} \quad \sum_{m=1}^{p} \boldsymbol{\theta}_{m}^{\top}((\mathbf{I}_{n} - \mathbf{H}\mathbf{H}^{\top}) \circ \mathbf{K}_{m})\boldsymbol{\theta}_{m}$$
$$\text{with respect to} \quad \boldsymbol{\Theta} \in \mathbb{R}_{+}^{n \times p} \qquad \text{(OPT8)}$$
$$\text{subject to} \quad \boldsymbol{\Theta}\mathbf{1}_{p} = \mathbf{1}_{n}$$

Training the localized combination approach requires more computational effort than training the global approach due to the increased size of QP problem in the second step. However, the block-diagonal structure of the Hessian matrix in (OPT8) can be exploited to solve this problem much more efficiently. Note that the objective function of (OPT8) can be written as

$$\begin{bmatrix} \boldsymbol{\theta}_{1} \\ \boldsymbol{\theta}_{2} \\ \vdots \\ \boldsymbol{\theta}_{p} \end{bmatrix}^{\top} \begin{bmatrix} (\mathbf{I}_{n} - \mathbf{H}\mathbf{H}^{\top}) \circ \mathbf{K}_{1} & \mathbf{0}_{n \times n} & \cdots & \mathbf{0}_{n \times n} \\ \mathbf{0}_{n \times n} & (\mathbf{I}_{n} - \mathbf{H}\mathbf{H}^{\top}) \circ \mathbf{K}_{2} & \cdots & \mathbf{0}_{n \times n} \\ \vdots & \vdots & \ddots & \vdots \\ \mathbf{0}_{n \times n} & \mathbf{0}_{n \times n} & \cdots & (\mathbf{I}_{n} - \mathbf{H}\mathbf{H}^{\top}) \circ \mathbf{K}_{p} \end{bmatrix} \begin{bmatrix} \boldsymbol{\theta}_{1} \\ \boldsymbol{\theta}_{2} \\ \vdots \\ \boldsymbol{\theta}_{p} \end{bmatrix},$$

where we have an $n \times n$ matrix for each kernel on the diagonal of the Hessian matrix.

## 5   Experiments

Clustering patients is one of the clinically important applications in cancer biology because it helps to identify prognostic cancer subtypes and to develop personalized strategies to guide therapy. Making use of multiple genomic characterizations in clustering is critical because different patients may manifest their disease in different genomic platforms due to cancer heterogeneity and measurement noise. We use the human colon and rectal cancer data set provided by TCGA consortium (The Cancer Genome Atlas Network, 2012), which contains several genomic characterizations of the patients, to test our new clustering algorithm in a challenging real-world application.

We use DNA copy number, mRNA gene expression, and DNA methylation data of the patients for clustering. In order to evaluate the clustering results, we use three commonly used clinical biomarkers of colon and rectal cancer (The Cancer Genome Atlas Network, 2012): (i) micro-satellite instability (i.e., a hypermutable phenotype caused by the loss of DNA mismatch repair activity) (ii) hypermutation (defined as having mutations in more than or equal to 300 genes), and (iii) mutation in BRAF gene. Note that these three biomarkers are not directly identifiable from the input data sources used. The preprocessed genomic characterizations of the patients can be downloaded from a public repository at `https://www.synapse.org/#!Synapse:syn300013`, where DNA copy number, mRNA gene expression, DNA methylation, and mutation data consist of 20313, 20530, 24980, and 14581 features, respectively. The micro-satellite instability data can be downloaded from `https://tcga-data.nci.nih.gov/tcga/dataAccessMatrix.htm`. In the resulting data set, there are 204 patients with available genomic and clinical biomarker data.

We implement kernel $k$-means clustering and its multiview variants in Matlab. Our implementations are publicly available at `https://github.com/mehmetgonen/lmkkmeans`. We solve the QP problems of the multiview variants using the Mosek optimization software (Mosek, 2014). For all methods, we perform 10 replications of $k$-means with different initializations as the last step and use the solution with the lowest sum-of-squares cost to decide cluster memberships.

We calculate four different kernels to use in our experiments: (i) $\mathbf{K}_\mathrm{C}$: the Gaussian kernel on DNA copy number data, (ii) $\mathbf{K}_\mathrm{G}$: the Gaussian kernel on mRNA gene expression data, (iii) $\mathbf{K}_\mathrm{M}$: the Gaussian kernel on DNA methylation data, and (vi) $\mathbf{K}_\mathrm{CGM}$: the Gaussian kernel on concatenated data (i.e., early combination). Before calculating each kernel, the input data is normalized to have zero mean and unit standard deviation (i.e., $z$-normalization for each feature). For each kernel, we set the kernel width parameter to the square root of the number of features in its corresponding view.

We compare seven clustering algorithms on this colon and rectal cancer data set: (i) kernel $k$-means clustering with $\mathbf{K}_\mathrm{C}$, (ii) kernel $k$-means clustering with $\mathbf{K}_\mathrm{G}$, (iii) kernel $k$-means clustering with $\mathbf{K}_\mathrm{M}$, (iv) kernel $k$-means clustering with $\mathbf{K}_\mathrm{CGM}$, (v) kernel $k$-means clustering with ($\mathbf{K}_\mathrm{C} + \mathbf{K}_\mathrm{G} + \mathbf{K}_\mathrm{M}$) / 3, (vi) multiple kernel $k$-means clustering with ($\mathbf{K}_\mathrm{C}$, $\mathbf{K}_\mathrm{G}$, $\mathbf{K}_\mathrm{M}$), and (vii) localized multiple kernel $k$-means clustering with ($\mathbf{K}_\mathrm{C}$, $\mathbf{K}_\mathrm{G}$, $\mathbf{K}_\mathrm{M}$). The first three algorithms are single-view clustering methods that work on a single genomic characterization. The fourth algorithm is the early integration approach that combines the views at the feature level. The fifth and sixth algorithms are intermediate integration approaches that combine the kernels using unweighted and weighted sums, respectively, where the latter is very similar to the formulations of Huang et al. (2012) and Yu et al. (2012). The last algorithm is our localized MKL approach that combines the kernels in a sample-specific way.

We assign three different binary labels to each sample as the ground truth using the three clinical biomarkers mentioned and evaluate the clustering results using three different performance metrics: (i) normalized mutual information (NMI), (ii) purity, and (iii) the Rand index (RI). We set the number of clusters to 2 for all of the algorithms because each ground truth label has only two categories.

We first show the kernel weights assigned to 204 colon and rectal cancer patients by our localized data fusion approach. As we can see from Figure 1, some of the patients are very well characterized by their DNA copy number data. Our localized algorithm assigns weights larger than 0.5 to DNA copy number data for most of the patients in the second cluster, whereas all three views are used with comparable weights for the remaining patients. Note that the kernel weights of each patient are strictly nonnegative and sum up to 1 (i.e., defined on the unit simplex). Our proposed clustering algorithm can identify the most informative genomic platforms in an unsupervised and patient-specific manner. Together with the better clustering performance and biological interpretation presented next, this particular application from cancer biology shows the potential for localized combination strategy.

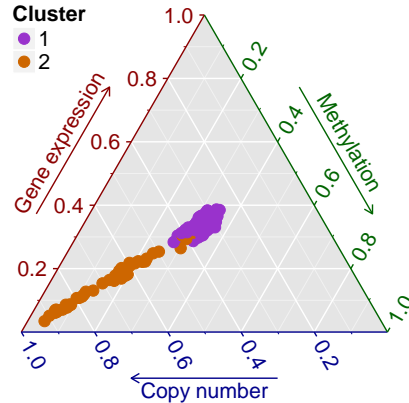

Figure 1: Kernel weights assigned to patients by our localized data fusion approach. Each dot denotes a single cancer patient, and patients in the same cluster are drawn with the same color.

Figure 2 summarizes the results obtained by seven clustering algorithms on the colon and rectal cancer data set. For each algorithm, the cluster assignment and the values of three clinical biomarkers are aligned to each other, and we report the performance values of nine biomarker–metric pairs. We see that DNA copy number (i.e., $\mathbf{K}_\mathrm{C}$) is the most informative genomic characterization when we compare the performance of single-view clustering algorithms, where it obtains better results than mRNA gene expression (i.e., $\mathbf{K}_\mathrm{G}$) and DNA methylation (i.e., $\mathbf{K}_\mathrm{M}$) in terms of NMI and RI on all biomarkers. We also see that the early integration strategy (i.e., $\mathbf{K}_\mathrm{CGM}$) does not improve the results because mRNA gene expression and DNA methylation dominate the clustering step due to the unsupervised nature of the problem. However, when we combine the kernels using an unweighted combination strategy, i.e., ($\mathbf{K}_\mathrm{C} + \mathbf{K}_\mathrm{G} + \mathbf{K}_\mathrm{M}$) / 3, the performance values are significantly improved compared to single-view clustering methods and early integration in terms of NMI and RI on all biomarkers. Instead of using an unweighted sum, we can optimize the combination weights using the multiple kernel $k$-means clustering of Section 3. In this case, the performance values are slightly improved compared to the unweighted sum in terms of NMI and RI on all biomarkers. Our localized data fusion approach significantly outperforms the other algorithms in terms of NMI and RI on "micro-satellite instability" and "hypermutation" biomarkers, and it is the only algorithm that can obtain purity values higher than the ratio of the majority class samples on "mutation in BRAF gene" biomarker. These results validate the benefit of our localized approach for the multiview setting.

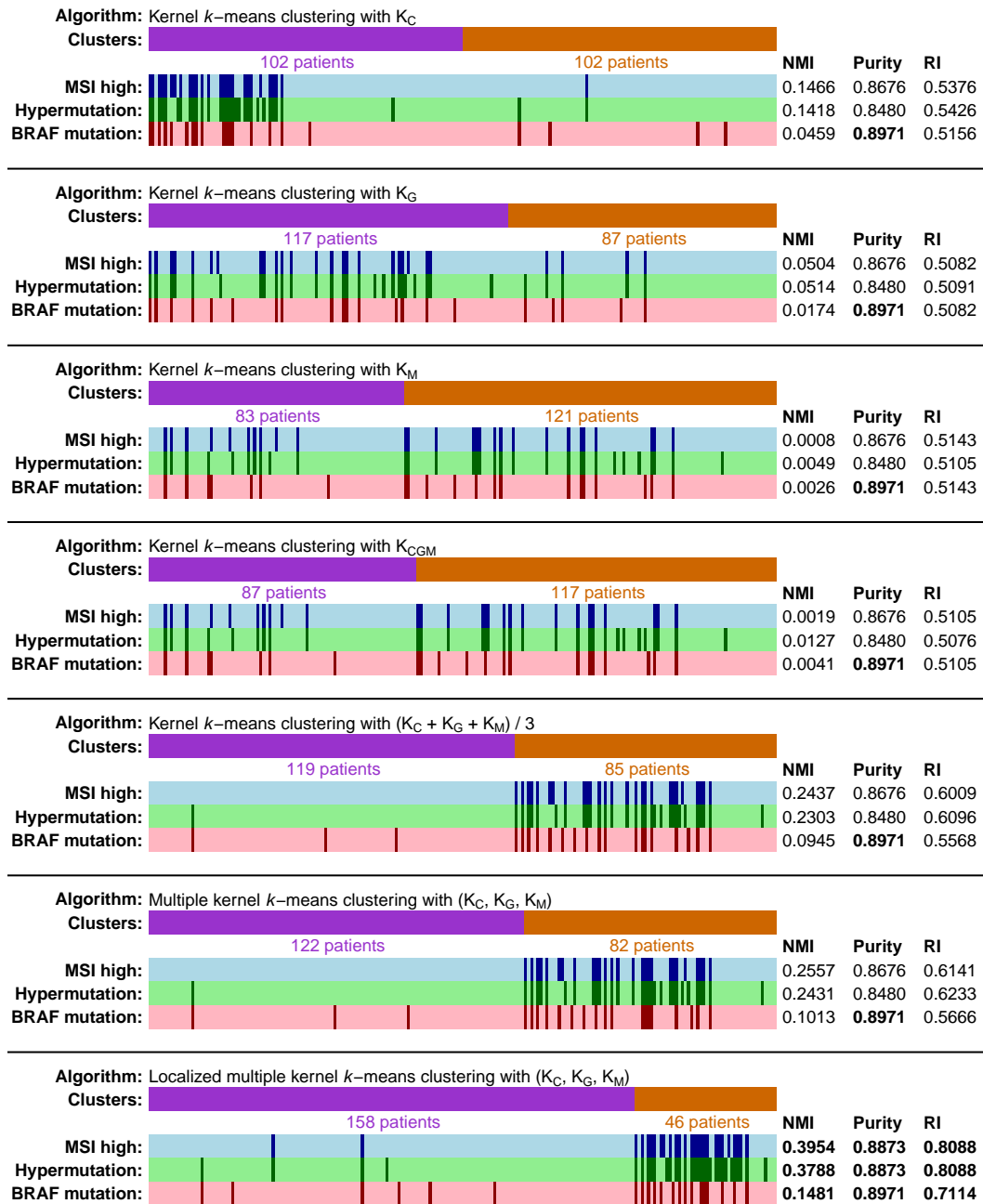

Figure 2: Results obtained by seven clustering algorithms on the colon and rectal cancer data set provided by TCGA consortium (The Cancer Genome Atlas Network, 2012). For each algorithm, we first display the cluster assignment and report the number of patients in each cluster. We then display the values of three clinical biomarkers aligned with the cluster assignment, where "MSI high" shows the patients with high micro-satellite instability status in darker color, "Hypermutation" shows the patients with mutations in more than or equal to 300 genes in darker color, and "BRAF mutation" shows the patients with a mutation in their BRAF gene in darker color. We compare the algorithms in terms of their clustering performance on three clinical biomarkers under three metrics: normalized mutual information (NMI), purity, and the Rand index (RI). For all performance metrics, a higher value means better performance, and for each biomarker–metric pair, the best result is reported in bold face. We see that our localized clustering algorithm obtains the best result for eight out of nine biomarker–metric pairs, whereas all algorithms have the same purity value for BRAF mutation.

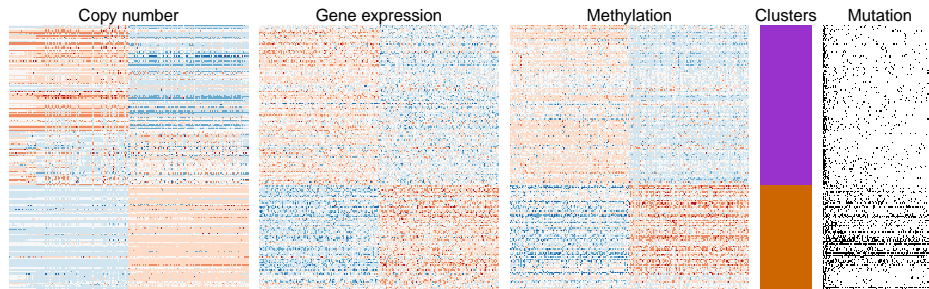

Figure 3: Important features in genomic views determined using the solution of multiple kernel $k$-means clustering together with cluster assignment and mutations in frequently mutated genes. For each genomic view, we calculate the Pearson correlation values between features and clustering assignment, and display topmost 100 positively correlated and bottommost 100 negatively correlated features (red: high, blue: low). We also display the mutation status (black: mutated, white: wild-type) of patients for 102 most frequently mutated genes, which are mutated in at least 16 patients.

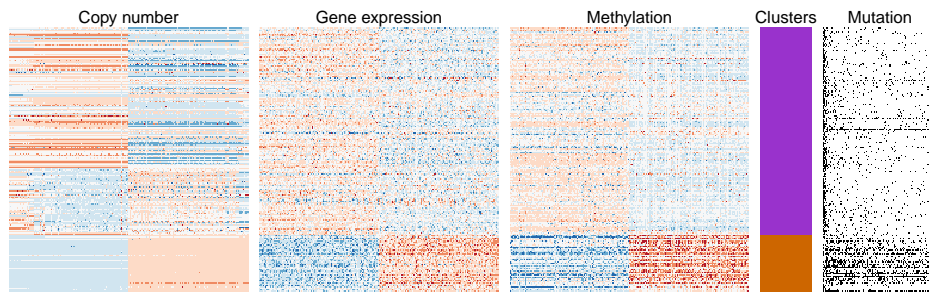

Figure 4: Important features in genomic views determined using the solution of localized multiple kernel $k$-means clustering together with cluster assignment and mutations in frequently mutated genes. See Figure 3 for details.

We perform an additional biological interpretation step by looking at the features that can be used to differentiate the clusters found. Figures 3 and 4 show features in genomic views that are highly (positively or negatively) correlated with the cluster assignments of the two best performing algorithms in terms of clustering performance, namely, multiple kernel $k$-means clustering and localized multiple kernel $k$-means clustering. We clearly see that the genomic signatures of the hyper-mutated cluster (especially the one for DNA copy number) obtained using our localized data fusion approach are much less noisy than those of global data fusion. Identifying clear genomic signatures are clinically important because they can be used for diagnostic and prognostic purposes on new patients.

## 6 Discussion

We introduce a localized data fusion approach for kernel $k$-means clustering to better capture sample-specific characteristics of the data in the multiview setting, which can not be captured using global data fusion strategies such as Huang et al. (2012) and Yu et al. (2012). The proposed method is from the family of MKL algorithms and combines the kernels defined on the views with sample-specific weights to determine the relative importance of the views for each sample. We illustrate the practical importance of the method on a human colon and rectal cancer data set by clustering patients using their three different genomic characterizations. The results show that our localized data fusion strategy can identify more relevant prognostic patient groups than global data fusion strategies.

The interesting topics for future research are: (i) exploiting the special structure of the Hessian matrix in our formulation by developing a customized solver instead of using an off-the-shelf optimization software to improve the time complexity, and (ii) integrating prior knowledge about the samples that we may have into our formulation to be able to find more relevant clusters.

**Acknowledgments.** This study was financially supported by the Integrative Cancer Biology Program (grant no 1U54CA149237) and the Cancer Target Discovery and Development (CTDD) Network (grant no 1U01CA176303) of the National Cancer Institute.

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
