[Reviews · NeurIPS 2014]

Submitted by Assigned_Reviewer_12

This paper generalizes multi-kernel k-means clustering (solved via relaxation) to the case where each clustered item (here, a person) gets an item-specific set of weights over the multiple kernels, rather than the traditional, shared, global weighting of the kernels. Using TCGA (cancer) data, with 3 modalities, they demonstrate that this generalization yields better clusterings than the traditional (global approach), when measured against 3 "bronze standard" clusterings arising from known clinical clusters.

The writing is clear, making for an easy read.

Although this is a somewhat incremental-seeming tweak, I think it was clever, with the potential to actually be used (rather than lost in the NIPS archives), and therefore of some significance.

Other comments:

In the introduction you mention that k-means is susceptible to local minima, and then use this to motivate the relaxation approach. However, as described on p3, even the relaxation approach ends up using k-means clustering, so this is a little puzzling to the reader.

Following on my previous comment, it would seem that your approach is sensitive to initialization, and susceptible to local minima. Please make clear if this is true or not, and if it is, please explain how you deal with this issue (in your, and the other algorithms) in the experimental section.

Also, why not use a naturally continuous-valued clustering like mixture of Gaussians, rather than the hard version of it of k-means?

On p. 5 you discuss "computational effort" and doing things "more efficiently". Please give the time complexity for the various algorithms being discussed so that the reader can appreciate these points. Similarly, you mention the notion of time complexity in the Discussion, but not any actual time complexities.

Following on the last point, what is the largest data set you expect to be able to handle with the current algorithm. 204 patients is rather small for current genomics data sets (and other data sets). As a point of reference, please give a timing for your experiments.

Concluding the success of the algorithm on the basis of one experiment, while not unusual for NIPS, is a little weak. Would be great if you could add in further experiments, not to mention some with a ground truth for objective comparison, if possible.

You use always only 2 clusters, but it would be interesting to know if your model is just as amenable to model selection (i.e. # of clusters) as other methods. Presumably so, but would be good to know.

p. 6 "aligned to each other"--slightly odd phrasing, but I eventually understood by looking at the figures. Similarly so the "nine biomarker-metric pairs".

p.7 would be good if you annotated the tie with non-bold (maybe with a *), as I was confused for a while about where the 8/9 things were, until I realized there was a tie.

Summary: Well-written paper, with nice experimental results (although in typical NIPS fashion, on only one data set), that may very well be actually used by people in the future.

Submitted by Assigned_Reviewer_30

This paper applies multiple kernel k-means clustering to multiview problems. Sample specific weights are learned as part of an intermediate integration of views approach and this is the main contribution of the paper. This is a localized weighting of the samples.

Quality: The paper is technically sound. Code and data are made available.

Clarity: Mostly good; I wasn't clear what the relaxing achieved in OPT4. (The authors mention in the rebuttal that binary Z is relaxed to real H.)

I'd like to see some mention of / comparison with model based clustering. I'd also like to see some discussion of the implicit assumptions made by k-kmeans such as equal sized clusters.

Originality: Good. I haven't seen locally specific sample weights used before.

Significance: Medium.
Summary: A method for learning sample specific weights for multiple kernel k-means clustering of multiview data is proposed. Algorithms for optimisation are detailed and the method is shown to compare favourably with existing related methods on a cancer biology dataset. Good paper, well written.

Submitted by Assigned_Reviewer_37

This manuscript describes a novel multiple kernel learning strategy
for clustering. The approach generalizes a previously described MKL
version of k-means clustering (Yu 2012), in which the relative weights
of the kernels are learned. The novelty of the current approach lies
in learning not only kernel-specific weights but also weights for each
example. The authors frame the resulting optimization problem and
solve it using an alternating approach.

The idea of doing MKL with sample-specific weights is not novel. See,
e.g., a 2006 ICML paper by Lewis et al. that does this in a supervised
setting. Hence, the primary novelty here lies in doing this kind of
learning in an unsupervised setting.

The experiments are nicely done, though I do have some suggestions.

o The authors should emphasize more clearly that the biomarkers used
to define the labels are not directly identifiable from any of the
three data sets used as input.

o In figure 1, the authors have apparently chosen to display one of
the three binary labels using color. Which one was chosen, and why?
This is not stated. Furthermore, the authors claim in the
accompanying text (and again below) that DNA copy number is the most
informative. Yet the figure suggests that DNA copy number and gene
expression are equally informative. It is just methylation that
seems to provide little information.

o The normalization described on line 274 needs to be clarified. Is
this row or column normalization?

o The results in Figure 2 are impressive. However, experiments should be done to
verify the robustness of these results to choice of hyperparameters.

o Figures 3 and 4 are only mildly informative. I agree with the
general goal of trying to see whether the learned example weights
are interesting, but it is not clear why the trend (less "noisy"
genomic signatures in the hyper-mutated genes) is useful.

Summary: This is a straightforward extension of an existing MKL method to learn
weights that are specific to each example. The empirical results are strong.

Author Feedback
Author rebuttal: ------------------------------------------------------------

Reviewer#12

* k-means in the last step and initialization in the experiments: After solving the relaxed problem (OPT4), one obtains the optimum H matrix, but this matrix should be converted into a cluster membership matrix. In order to achieve this, rows of H are normalized to be on the unit sphere, and clustering solution can be extracted by performing k-means on the full matrix (also known as spectral clustering). Note that, after the normalization step, H matrix contains k-dimensional representations of the samples on the unit sphere, and k-means is not very sensitive to initialization in such a case. We will make this more clear.

For all algorithms we compared, we performed 10 replications of k-means with different initializations as the last step after solving (OPT4) and picked the solution with the lowest sum-of-squares cost function. We will explicitly mention these points in the manuscript, and thanks for suggesting this.

* Hard clustering: In our study, we preferred to perform hard clustering instead of soft clustering, which would bring additional interpretation difficulties in clinical settings.

* Computational complexity: We did not report the running times explicitly due to space constraints. However, our method takes approximately 4 seconds on a standard laptop for colon and rectal cancer data set (204 patients), whereas other algorithms take approximately 1 second. We used an off-the-shelf optimization software (i.e., Mosek) to solve underlying QP problems. Even with this approach, current implementation is scalable to data sets with a few thousand patients. As we mentioned in the manuscript, the algorithm can even be made faster using a customized optimizer that exploits the special structure of the Hessian matrix. We will explicitly mention these points in the manuscript, and thanks for suggesting this.

* Additional experiments on other data sets: We were planning to perform additional experiments on other cancer data sets, but colon and rectal cancer data set has clearly defined clinical biomarkers, which enable an objective comparison of results.

* Data sets with ground truth: Unfortunately, in clinical applications, it is not very easy to find data sets with gold standard labelings. That is why we used a single data set with clinically accepted biomarkers to compare the algorithms as objectively as possible.

* Number of clusters: We performed experiments with k = 2, 3, and 4, but we chose to report the results with k = 2 due to the binary nature of clinical biomarkers used to evaluate the results. However, comparison between algorithms with k = 3 and 4 produced similar conclusions in terms of performance metrics used.

------------------------------------------------------------

Reviewer#30

* Relaxation in (OPT4): We will explicitly mention that binary Z matrix is relaxed to real-valued H matrix.

------------------------------------------------------------

Reviewer#37

* Clinical biomarkers: Good point! As you mentioned, the biomarkers we used are not directly identifiable from three input data sources. We will make this more clear, and thanks for suggesting this.

* Interpretation of Figure 1: In this figure, clinical biomarkers were not displayed at all. Node colors represent the cluster assignment (purple: cluster#1, brown: cluster#2) obtained using our method. Note that gene expression and methylation are assigned weights less than 0.4 for all patients, whereas copy number has weights between 0.3 and 1.0. For example, if we look at the kernel weights assigned to cluster#2 (i.e., brown nodes), we see that copy number has weights larger than 0.5, whereas gene expression and methylation have weights less than 0.25. That is why we concluded copy number is the most informative data source.

* Data normalization: Data sources were normalized over columns (i.e., features) to have zero mean and unit standard deviation. We will make this more explicit.

* Hyper-parameter choice: This was discussed in Lines 274--275. Note that, due to the unsupervised nature of the problem, it is not trivial to perform hyper-parameter optimization. That is why we set the kernel width parameter to the mean of pairwise Euclidean distances between the samples for each kernel, which is a standard heuristic frequently used in the literature. However, we obtained similar results with other kernel parameters.

* Interpretation of Figures 3 and 4: We reported these figures because identifying clear signatures for subtypes in a clinical setting is diagnostically and prognostically important. These signatures, which consist of hundreds of genes, can be used for new patients without performing these genomic screens in full, leading to reduced cost and time.

------------------------------------------------------------

Meta-reviewer

* Yu et al. citation: Thanks for pointing out the missing authors of this citation. We will fix this issue.

* Difference from biclustering methods: Biclustering algorithms try to group samples (e.g., patients) and features (e.g., genomic markers) simultaneously, which produces clusters of samples associated with some of the features. However, we have multiple views of the samples and are only interested in clustering over them using all of the views. While doing this, our method also identifies which data sources (i.e., views) are informative for each patient by assigning localized weights to capture sample-specific characteristics of the data, whereas biclustering algorithms pick the same features for the samples within each cluster and do not assign any weights to features.

------------------------------------------------------------